# The Risk of Depression in Patients with Pemphigus: A Nationwide Cohort Study in Taiwan

**DOI:** 10.3390/ijerph17061983

**Published:** 2020-03-17

**Authors:** Yi-Min Hsu, Hsin-Yu Fang, Cheng-Li Lin, Shwn-Huey Shieh

**Affiliations:** 1Department of Public Health, China Medical University, Taichung 40447, Taiwan; n4006@mail.cmuh.org.tw; 2Department of Nursing, China Medical University Hospital, Taichung 40447, Taiwan; 3Department of dermatology, China Medical University Hospital, Taichung 40447, Taiwan; 4Management Office for Health Data, China Medical University Hospital, Taichung 40447, Taiwan; A21467@mail.cmuh.org.tw; 5Department of Health Services Administration, China Medical University, Taichung 40447, Taiwan; 6Department of Nursing, Asia University, Taichung 41354, Taiwan

**Keywords:** depression, pemphigus, cohort study

## Abstract

Pemphigus is a chronic dermatological disorder caused by an autoimmune response and is associated with a high proportion of comorbidities and fatalities. The aim of this study was to investigate the risk of depression in patients with pemphigus. Data were derived from the National Health Insurance Research Database recorded during the period 2000–2010 in Taiwan. Multivariate Cox proportional hazards regression models were used to analyze the data and assess the effects of pemphigus on the risk of depression after adjusting for demographic characteristics and comorbidities. Patients with pemphigus were 1.98 times more likely to suffer from depression than the control group (pemphigus, adjusted HR: 1.99, 95% CI = 1.37–2.86). People aged ≥65 years were 1.69 times more likely to suffer from depression than those aged 20–49 years (≥65 years, adjusted HR: 1.42, 95% CI = 0.92–2.21). Female and male patients with pemphigus were respectively 2.02 and 1.91 times more likely to suffer from depression than the control group (female, adjusted HR: 2.09, 95% CI = 1.24–3.54; male, adjusted HR: 1.87, 95% CI = 0.97–3.60). People with HTN, hyperlipidemia, asthma/COPD, and chronic liver disease were respectively 1.73, 2.3, 2.2, and 1.69 times more likely to suffer from depression than those without these comorbidities (HTN, adjusted HR: 0.75, 95% CI = 0.41–1.42; hyperlipidemia, adjusted HR: 1.48, 95% CI = 0.78–2.82; asthma/COPD, adjusted HR: 1.4, 95% CI = 0.72–2.69; and chronic liver disease, adjusted HR: 1.61, 95% CI = 1.07–2.43). There was a significant association between pemphigus and increased risk of depression. Female patients had a higher incidence of depression.

## 1. Introduction

Pemphigus is an autoimmune disease characterized by intraepithelial bullae and erosions in the skin and mucosa. The skin lesions in patients with pemphigus can be pruritic and painful, and the erosions can be extended. Some studies have reported that the incidence of pemphigus was 0.076 and 1.6 per 100,000 person-years [1,2]. The risk of developing pemphigus is higher in Female and elderly patients [1,3]. An earlier study also reported that pemphigus is associated with a higher risk of death [3]. Due to the difficulty in treating this disease, the hospital stays are long, requiring immunosuppressive drugs, and the recurrence rate is also high, which probably has a negative impact on the patient’s appearance. Patients with pemphigus experience a decreased quality of life that affects their physical, psychological, and social aspects [4,5].

According to statistics from the World Health Organization, there are approximately 350 million people suffering from depression across the world. It has been estimated that by the year 2020, depression will be the second leading cause of human disability. Thus, depression is obviously one of the world’s important public health issues. Depression can not only cause suffering in human beings, but it also increases the social and economic costs. The risk factors for depression include female gender, being unemployed, low education status, and diseases that significantly affect daily life [6]. Several chronic conditions have been associated with depression, such as infectious diseases [7], respiratory diseases [8], cardiovascular diseases [9], and cancers [10].

Depression is also a common mental illness in people with dermatological disorders [11,12]. Approximately 10%–17% of patients with dermatological disorders are frequently associated with psychiatric comorbidities. These mood disorders are generally underestimated by dermatologists [13]. A previous study demonstrated that pemphigus may increase the risk of developing psychiatric disorders and affect mental health [14]. When the symptoms of pemphigus and depressive mood are combined, it may result in serious physical and mental burden on the patient. Hence, physicians should pay more attention to this issue. We investigated whether there was an increase in the risk of depression in patients with pemphigus using a retrospective nationwide cohort study design and data retrieved from the Taiwan National Health Insurance Research Database (NHIRD).

## 2. Methods

### 2.1. Data Source

Data were derived from the National Health Insurance Research Database (NHIRD). This registry contains the data of approximately 23.75 million individuals. All the registration and claims data of these 1,000,000 individuals collected by the National Health Insurance program constitute the LHID2000. There was no significant difference in gender distribution (chi-square = 1.74, df = 1, *p* = 0.187) between the patients in the LHID2000 and the original NHIRD. Patients’ diagnoses in the database were encoded using the International Classification of Diseases, 9th Revision, Clinical Modification (ICD-9-CM) code. To protect personal privacy, any information in the database leading to identity exposure was encrypted before being sent to the National Health Research Institutes for database construction and is further encrypted before being released to each researcher. The Research Ethics Committee of the China Medical University approved this research.

### 2.2. Subject Selection

Data of subjects aged ≥20 years were selected from NHIRD in our 2000–2010 study. First, we discussed whether there is any distinction between the pemphigus (ICD-9-CM code 694.4) and nonpemphigus (ICD-9-CM code 694.4) cohorts in terms of comorbidities, including hypertension (HTN) (ICD-9-CM codes 401–405), diabetes mellitus (DM) (ICD-9-CM code 250), hyperlipidemia (ICD-9-CM code 272), asthma/chronic obstructive pulmonary disease (COPD) (ICD-9-CM code 493, 496), chronic liver disease and cirrhosis (ICD-9-CM code 571), chronic kidney disease (CKD) (ICD-9-CM code 585), and cancer (ICD-9-CM code 140-208). The index date was the date of an initial diagnosis of pemphigus. History of depression before the index date and any missing medical information were excluded in both cohorts. Using the frequency matching method, we randomly sampled 926 patients with pemphigus and 3674 people without pemphigus such that the proportion of people with pemphigus to the nonpemphigus cohort in the index years and the strata of age group, sex, monthly income, urbanization level, and occupation was approximately 1:4. People in both cohorts were followed up, and the person-years of follow-up duration were estimated until one of the following conditions was met: the incidence of depression was diagnosed, censored for death, withdrawal from the insurance, or censored by the end of 2011.

### 2.3. Statistical Analysis

First, we conducted Student’s *t*-test for age and mean ± SD (year) and chi-squared test for age group (year), sex, monthly income, urbanization level, occupation, and strata of comorbidity. Second, we fit details about pemphigus, age, year, gender, monthly income, urbanization level, occupation, and comorbidity with univariable Cox proportional hazards regression models to derive hazard ratios and conducted a multivariable Cox proportional hazards regression analysis on those covariates revealing significant crude HRs to derive adjusted HRs. In a similar manner, we separated the pemphigus and nonpemphigus cohorts to derive hazard ratios of the two cohorts in terms of age group, sex, monthly income, urbanization level, occupation category, comorbidity, and follow-up. Finally, we used the Kaplan–Meier method and verified the difference between the two cohorts through a log-rank test. All statistical analyses were conducted using the SAS statistical package (version 9.4 for Windows; SAS Institute, Inc, Cary, NC, USA), and *p* < 0.05 was accepted for statistical significance.

## 3. Results

As shown in Table 1, the mean ± SD ages of the pemphigus and nonpemphigus cohorts were respectively 52.6 ± 16.5 and 52.9 ± 16.6 years (*p* = 0.58). Patients with pemphigus were more likely to have HTN, DM, hyperlipidemia, and asthma/COPD than people without pemphigus (all *p* values < 0.001). Patients in the pemphigus cohort were not likely to have chronic liver disease, CKD, and cancer when compared with the nonpemphigus cohort (chronic liver disease, *p* = 0.93; CKD, *p* = 0.88; cancer, *p* = 0.84).

Table 2 demonstrates that the pemphigus cohort was 1.98 times more likely to suffer from depression than the control group (pemphigus, adjusted HR: 1.99, 95% CI = 1.37–2.86). People aged ≥65 years were 1.69 times more likely to suffer from depression than people aged 20–49 years (≥65 years, adjusted HR: 1.42, 95% CI = 0.92–2.21). Those with monthly income (New Taiwan Dollar, NTD) <15,000 were 2.05 times more likely to suffer from depression than those with monthly income ≥20,000 (<15,000, adjusted HR: 1.81, 95% CI = 1.07–3.06). Those who were neither office workers nor laborers were 1.69 times more likely to have depression than those who were office workers (others, adjusted HR: 1.10, 95% CI = 0.61–1.98). People who had HTN, hyperlipidemia, asthma/COPD, and chronic liver disease were respectively 1.73, 2.3, 2.2, and 1.69 times more likely to suffer from depression than those did not have these comorbidities (HTN, adjusted HR: 0.75, 95% CI = 0.41–1.42; hyperlipidemia, adjusted HR: 1.48, 95% CI = 0.78–2.82; asthma/COPD, adjusted HR: 1.4, 95% CI = 0.72–2.69; chronic liver disease, adjusted HR: 1.61, 95% CI = 1.07–2.43).

However, after conducting the multivariable Cox proportional hazards regression analysis for age, monthly income, occupation, HTN, hyperlipidemia, asthma/COPD, and chronic liver disease, it was found that people aged ≥65 years were not more likely to have depression than people aged 20–49 years, considering other covariates to be constant. Holding other covariates constant, the results showed that those who were neither office workers nor laborers were not more likely to suffer from depression than those who were office workers. Considering other covariates to be constant, those with HTN, HL, and asthma/COPD were not more likely to have depression than those without these comorbidities. Medications with corticosteroid and azathioprine showed no significantly increased the risk of depression.

As shown in Table 3, patients in the pemphigus cohort aged 20–49 and 50–64 years were respectively 2.07 and 2.41 times more likely to have depression than the control group (20–49 years, adjusted HR: 1.86, 95% CI = 1.06–3.26; 50–64 years, adjusted HR: 1.98, 95% CI = 0.75–5.26). Female and male patients with pemphigus were respectively 2.02 and 1.91 times more likely to suffer from depression than the control group (females, adjusted HR: 2.09, 95% CI = 1.24–3.54; males, adjusted HR: 1.87, 95% CI = 0.97–3.60). In terms of monthly income, patients with pemphigus with <15,000 and ≥20,000 were respectively 2.42 and 2.16 times more likely to have depression than the control population (<15,000, adjusted HR: 2.28, 95% CI = 1.14–4.58; ≥20,000, adjusted HR: 2.29, 95% CI = 1.03–5.10). Regarding the urbanization level two people, the pemphigus cohort was 2.59 times more likely to have depression than the nonpemphigus cohort (urbanization level 2, adjusted HR: 1.73, 95% CI = 0.79–3.76). In terms of office workers, the pemphigus cohort was 2.53 times more likely to suffer from depression than the nonpemphigus cohort (office workers, adjusted HR: 2.49, 95% CI = 1.41–4.39). Among people without comorbidities, the pemphigus cohort was 1.87 times more likely to suffer from depression than the control population (no comorbidity, adjusted HR: 2.13, 95% CI = 1.28–3.57). Furthermore, patients in the pemphigus cohort who were followed up for <6 and ≥6 months were respectively 5.23 and 1.69 times more likely to suffer from depression than the nonpemphigus cohort (<6 months, adjusted HR: 4.33, 95% CI = 1.42–13.2; ≥6 months, adjusted HR: 1.76, 95% CI = 1.121–2.75).

However, after performing the multivariable Cox proportional hazards regression analysis for age, monthly income, occupation, HTN, HL, asthma/COPD, and CLD, the pemphigus cohort showed no significant increase in the hazards for depression compared with the nonpemphigus cohort in terms of age group 50–64 years, male gender, and urbanization level 2, considering other covariates to be constant. In terms of urbanization level 1, the pemphigus cohort was 2.04 times more likely to suffer from depression than the nonpemphigus cohort, considering other covariates to be constant (urbanization level 1, adjusted HR: 2.04, 95% CI = 1.05–3.94.) (Figure 1).

## 4. Discussion

In this nationwide population-based cohort study, we observed that during a mean follow-up period of 10.9 years, the females with asthma in the pemphigus cohort had a 1.98-fold higher risk of depression compared with those without pemphigus. Although HTN, hyperlipidemia, asthma/COPD, and chronic liver disease were associated with a higher risk of depression, the risk of depression remained significantly higher in patients with pemphigus after adjustment for these covariates.

Pemphigus vulgaris is a rare disease and its geographic distribution is more variable [15]. Pemphigus affects both mucous membranes and the skin and Pemphigus vulgaris and pemphigus foliaceus are the two major subtypes of pemphigus. Atypical pemphigus can present with variable clinical manifestations and histological result [16]. It is easily recurrent disease and can potentially impact the quality of life. The pathogenic relevance of such autoantibodies has been largely demonstrated experimentally [17,18,19]. The first-line therapy comprises systemic corticosteroids and other therapies, including immunosuppressive agents or immunoglobulins [19,20]. Several studies have reported that dermatological patients have psychiatric problems such as anxiety neurosis and neurotic depression [12,21,22]. Wohl et al. demonstrated an increased rate of depression in patients with pemphigus in a case–control study [5].

A previous study demonstrated that dermatologists frequently underestimated the presence of mood disorders in patients with significant skin diseases [13]. In fact, there is extensive research indicating an increased risk of depression in patients with dermatological disorders such as psoriasis, alopecia areata, and atopic dermatitis [23,24,25]. Pemphigus may have a negative impact on the patients’ body image [26]. It may be frequently recurrent and not easy to treat. In the present study, we observed an increased prevalence of depression in patients with pemphigus. Depression could have a negatively effect on the quality of life and sometimes it becomes a serious illness. However, depression is treatable. Therefore, besides treating pemphigus in patients, physicians should also focus on the situation in which the patient may develop depression.

Regarding the relationship between age and risk of depression in patients with pemphigus, a previous research conducted in Germany in 1999 reported that elderly patients had a higher risk of developing pemphigus. It was found that patients aged 61–70 and 81–90 years respectively had 15- and 164-times higher risk than people aged 1–60 years. Hence, the incidence of disease increases with age [27]. In our study, the risk of depression was found to be higher in patients aged <49 years. It is speculated that the symptoms caused by the disease and the mental image of patients with pemphigus can easily affect their daily activities, work, and psychology. Therefore, young-age patients perhaps had a higher incidence of depression. Wohl et al. demonstrated the risk of pemphigus and the risk of depression, and was consistent with the most significant result of age less than 40 years [5].

Pemphigus patients experienced more impaired quality of life [28]. Previous study demonstrated a strong impact of pemphigus on health status and quality of life, especially in women [29]. Regarding gender, we found the females had a significantly higher risk of depression than men. This result was consistent with the result of a study conducted for 20 years in Taiwan [6]. Wohl et al. investigated the relationship between pemphigus and the risk of depression and found that patients with pemphigus have a high risk of developing depression (OR = 1.19; 95% CI: 1.12–1.27, *p* < 0.001) [5]. They found a higher risk factor of depression in female patients. This result is the same with the results of our study, which may be associated with the risk that female patients are more likely to be affected by the disease caused by the disease cannon rash. It is not known whether this is caused by differences in ethnicity [30]. Further research may be considered to investigate this aspect.

Based on our review of the relevant literature, this is the first cohort study to suggest the females with pemphigus are at an increased risk of depression. However, several limitations must be considered in this study. First, the NHIRD does not provide detailed personal information of enrollees, such as the severity of diseases and the lifestyle of patients, which are apparent potential risk factors for depression. Second, the diagnoses of diseases were based on ICD-9-CM codes obtained from the nationwide administrative database; participants who did not seek medical care were not included in the study. We also can’t identify if the depression improved after pemphigus control or stop corticosteroid use. Third, we can’t exactly and accurately differentiate the types of pemphigus, such as pemphigus vulgaris or pemphigus foliaceus in the NHIRD because there is no information available from the diagnostic codes of ICD-9-CM. Fourth, all data in the NHIRD are anonymous. Thus, relevant clinical variables, such as clinical symptoms/signs or questionnaire were unavailable for our study cases. However, considering that the accessibility and availability of medical care within the NHI are high in Taiwan [31], we believe that any underestimation was very limited. Despite these limitations, we believe that the relationship between pemphigus and depression found in this study is reliable and applicable to the general population because of the large-scale, nationwide, population-based study design, validity of the database, and the 11-year follow-up period.

## Figures and Tables

**Figure 1 ijerph-17-01983-f001:**
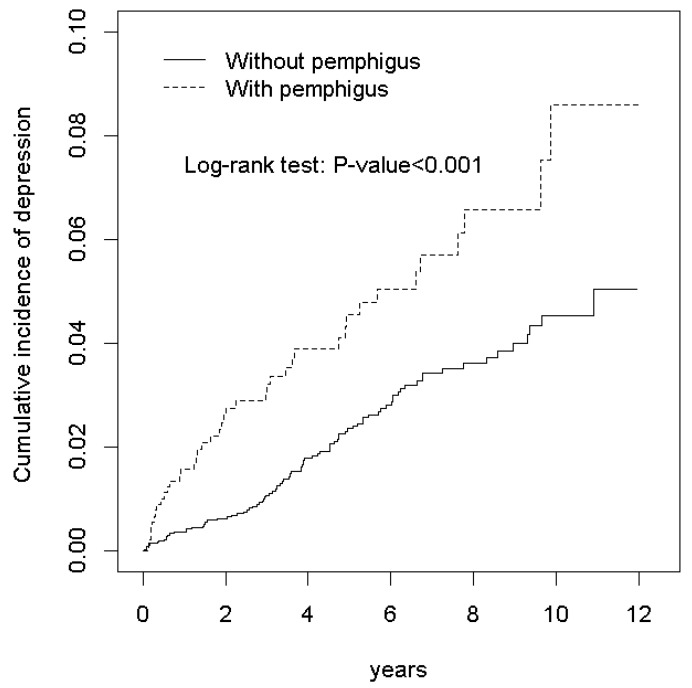
Cumulative incidence of depression between individuals with and without pemphigs.

**Table 1 ijerph-17-01983-t001:** Baseline characteristics in individuals with and without pemphigus.

Characteristics	Pemphigus		*p*-Value
	No	Yes	
	N = 3674	N = 926	
	n (%)	n (%)	
Age group (year)			0.92
20−49	1768 (48.1)	442 (47.7)	
50−64	1012 (27.6)	253 (27.3)	
≥65	894 (24.3)	231 (25.0)	
Age, mean ± SD ^a^ (year)	52.6 ± 16.5	52.9 ± 16.6	0.58
Sex			0.97
Women	1956 (53.2)	493 (53.2)	
Men	1718 (46.8)	433 (46.8)	
Monthly income ^b^			0.99
<15,000	819 (22.3)	206 (22.3)	
15,000−19,999	1663 (45.3)	418 (45.1)	
≥20,000	1192 (32.4)	302 (32.6)	
Urbanization level ^c^			0.99
1 (highest)	1197 (32.6)	302 (32.6)	
2	1141 (31.1)	288 (31.1)	
3	592 (16.1)	149 (16.1)	
4 (lowest)	744 (20.3)	187 (20.2)	
Occupation ^d^			0.99
Office worker	2058 (56.0)	519 (56.1)	
Laborer	1291 (35.1)	325 (35.1)	
Other	325 (8.85)	82 (8.86)	
Comorbidity			
HTN	204 (5.55)	282 (30.5)	<0.001
DM	75 (2.04)	93 (10.0)	<0.001
HL	100 (2.72)	180 (19.4)	<0.001
Asthma/COPD	143 (3.89)	82 (8.86)	<0.001
CLD	642 (17.5)	163 (17.6)	0.93
CKD	58 (1.58)	14 (1.51)	0.88
Cancer	91 (2.48)	24 (2.59)	0.84
Medicines			
Steroids	3404 (92.7)	926 (100.0)	<0.001
Azathioprin	3 (0.08)	423 (45.7)	<0.001

HTN, hypertension; DM, diabetes mellitus; HL, hyperlipidemia; COPD, chronic obstructive pulmonary disease; CLD, chronic liver disease and cirrhosis; CKD, chronic kidney disease; SD, standard deviation; Chi-square test; ^a^
*t*-test; ^b^ Unit: New Taiwan dollar (NTD), 1 NTD is equal to 0.03 US dollar; ^c^ Urbanization level was categorized according to the population density of the residential area into 4 levels, with Level 1 the most urbanized and Level 4 the least urbanized; ^d^ Other occupation category included primarily retired, unemployed, and low-income groups.

**Table 2 ijerph-17-01983-t002:** The incidence and risk factors for depression.

	Event	PY	Rate ^#^	Crude HR (95% CI)	Adjusted HR ^&^ (95% CI)
Pemphigus					
No	91	20,282	4.49	1.00	1.00
Yes	41	4606	8.90	1.98 (1.37, 2.86) ***	1.99 (1.32, 3.00) **
Age, year					
20−49	63	12,921	4.88	1.00	1.00
50−64	25	6656	3.76	0.77 (0.48, 1.22)	0.71 (0.45, 1.14)
≥65	44	5311	8.28	1.69 (1.15, 2.49) **	1.42 (0.92, 2.21)
Gender					
Women	78	13,893	5.61	1.00	
Men	54	10,995	4.91	0.87 (0.62, 1.23)	
Monthly income ^a^					
<15,000	42	5408	7.77	2.05 (1.29, 3.24) **	1.81 (1.07, 3.06) *
15,000−19,999	58	11,028	5.26	1.39 (0.90, 2.14)	1.17 (0.71, 1.93)
≥20,000	32	8452	3.79	1.00	1.00
Urbanization level ^b^					
1 (highest)	47	8435	5.57	1.00	
2	38	7570	5.02	0.90 (0.59, 1.38)	
3	18	3871	4.65	0.83 (0.48, 1.44)	
4 (lowest)	29	5013	5.79	1.04 (0.65, 1.65)	
Occupation ^c^					
Office worker	66	14,034	4.70	1.00	1.00
Laborer	48	8594	5.59	1.19 (0.82, 1.72)	1.21 (0.76, 1.93)
Other	18	2260	7.97	1.69 (1.01, 2.85) *	1.10 (0.61, 1.98)
Comorbidity					
HTN					
No	114	22,817	5.00	1.00	1.00
Yes	18	2071	8.69	1.73 (1.05, 2.85) *	0.75 (0.41, 1.42)
DM					
No	126	24,163	5.21	1.00	1.00
Yes	6	725	8.27	1.59 (0.70, 3.58)	
HL					
No	117	23,580	4.96	1.00	1.00
Yes	15	1308	11.5	2.30 (1.34, 3.94) **	1.48 (0.78, 2.82)
Asthma/COPD					
No	121	23,912	5.06	1.00	1.00
Yes	11	976	11.3	2.20 (1.19, 4.09) *	1.40 (0.72, 2.69)
CLD					
No	99	20,806	4.76	1.00	1.00
Yes	33	4082	8.08	1.69 (1.14, 2.51) **	1.61 (1.07, 2.43) *
CKD					
No	130	24,611	5.28	1.00	1.00
Yes	2	277	7.21	1.36 (0.34, 5.51)	
Cancer					
No	129	24,456	5.27	1.00	1.00
Yes	3	432	6.95	1.30 (0.41, 4.08)	
Medicines Steroids					
No	3	1313	2.28	1.00	1.00
Yes	129	23,575	5.47	2.39 (0.76, 7.50)	
Azathioprin					
No	115	22,784	5.05	1.00	1.00
Yes	17	2104	8.08	1.60 (0.96, 2.67)	

CI, confidence interval; HR, hazard ratio; PY, person-years; HTN, hypertension; DM, diabetes mellitus; HL, hyperlipidemia; COPD, chronic obstructive pulmonary disease; CLD, chronic liver disease and cirrhosis; CKD, chronic kidney disease; Rate ^#^, incidence rate per 1000 person-years; ^&^ Multivariable analysis including age, monthly income, occupation, comorbidities of HTN, HL, asthma/COPD, and CLD; * *p* < 0.05, ** *p* < 0.01, *** *p* < 0.001. ^a^ Unit: New Taiwan dollar (NTD), 1 NTD is equal to 0.03 US dollar; ^b^ Urbanization level was categorized according to the population density of the residential area into 4 levels, with Level 1 the most urbanized and Level 4 the least urbanized; ^c^ Other occupation category included primarily retired, unemployed, and low-income groups.

**Table 3 ijerph-17-01983-t003:** Incidences and hazard ratios of depression between individuals with and without pemphigus.

	Pemphigus							
	No			Yes				
	Event	PY	Rate ^a^	Event	PY	Rate ^a^	Crude HR (95% CI)	Adjusted HR ^b^ (95% CI)
Age group								
20−49	42	10,411	4.03	21	2510	8.37	2.07 (1.23, 3.49) **	1.86 (1.06, 3.26) *
50−64	16	5405	2.96	9	1251	7.20	2.41 (1.06, 5.45) *	1.98 (0.75, 5.26)
≥65	33	4466	7.39	11	845	13.0	1.75 (0.88, 3.47)	1.709 (0.75, 3.87)
Sex								
Female	53	11,264	4.71	25	2629	9.51	2.02 (1.26, 3.25) **	2.09 (1.24, 3.54) **
Male	38	9018	4.21	16	1977	8.09	1.91 (1.07, 3.43) *	1.87 (0.97, 3.60)
Monthly income ^c^								
<15,000	28	4484	6.24	14	925	15.1	2.42 (1.27, 4.60) **	2.28 (1.14, 4.58) *
15,000−19,999	42	8987	4.67	16	2041	7.84	1.68 (0.94, 2.99)	1.57 (0.81, 3.04)
≥20,000	21	6811	3.08	11	1640	6.71	2.16 (1.04, 4.49) *	2.29 (1.03, 5.10) *
Urbanization level ^d^								
1 (highest)	33	6846	4.82	14	1589	8.81	1.81 (0.97, 3.38)	2.04 (1.05, 3.94) *
2	24	6167	3.89	14	1403	9.98	2.59 (1.34, 5.01) **	1.73 (0.79, 3.76)
3	13	3160	4.11	5	711	7.04	1.69 (0.60, 4.73)	1.85 (0.62, 5.52)
4 (lowest)	21	4109	5.11	8	903	8.86	1.70 (0.75, 3.83)	2.29 (0.90, 5.81)
Occupation category ^e^								
Office worker	42	11,441	3.67	24	2593	9.25	2.53 (1.53, 4.17) ***	2.49 (1.41, 4.39) **
Laborer	36	6998	5.14	12	1596	7.52	1.46 (0.76, 2.81)	1.38 (0.66, 2.88)
Other	13	1843	7.05	5	417	12.0	1.71 (0.61, 4.80)	1.68 (0.55, 5.09)
Comorbidity ^f^								
No	61	15,770	3.87	20	2765	7.23	1.87 (1.13, 3.11) *	2.13 (1.28, 3.57) **
Yes	30	4512	6.65	21	1841	11.4	1.709 (0.97, 2.97)	1.73 (0.99, 3.03)
Follow-up								
<6 months	7	1825	3.83	9	447	20.1	5.23 (1.95, 14.0) **	4.33 (1.42, 13.2) *
≥6 months	84	18,457	4.55	32	4159	7.69	1.69 (1.13, 2.54) *	1.76 (1.121, 2.75) *

^a^ Incidence rate per 1000 person-years; ^b^ Multivariable analysis including age, monthly income, occupation, comorbidities of HTN, HL, asthma/COPD, and CLD; ^c^ Unit: New Taiwan dollar (NTD), 1 NTD is equal to 0.03 US dollar; ^d^ Urbanization level was categorized according to the population density of the residential area into 4 levels, with Level 1 the most urbanized and Level 4 the least urbanized; ^e^ Other occupation category included primarily retired, unemployed, and low-income groups; ^f^ Individuals with any comorbidity of HTN, IHD, CVD, COPD, PUD, CLD, CKD, DM, gout, and RA/Diffuse diseases of connective tissue were classified into the comorbidity group; * *p* < 0.05, ** *p* < 0.01, *** *p* < 0.001.

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
