# Peer review of "The Risk of Depression in Patients with Pemphigus: A Nationwide Cohort Study in Taiwan"

_ijerph, 2020, doi:10.3390/ijerph17061983_

Round 1
Reviewer 1 Report
THE PAPER CAN NOW BE PUBLISHED
This manuscript is a resubmission of an earlier submission. The following is a list of the peer review reports and author responses from that submission.
Round 1
Reviewer 1 Report
This study compared rates based on helthcare coding in Taiwan and found increased rates if depression and other diseases in the pemphigus group, especially the females.
Since glucocorticosteroids are the mainstay of treatment for pemphigus and many if the itger diseases that were more prevalent in P pts, how can you distinguish the proportion of this due to the GCDs? Does this database record treatments?
If so, after GCDs are stopped etc. does the depression improve?
Female in the discussion should say ‘the females’..
Please cite work by Jain S et al 2017 about body image and blistering diseases.
Several QOL studies in pemphigus showed worse QOL in female pts- cite these.
Reviewer 2 Report
In the literature a clear correlation between the disease Pemphigus and depression of the patients has been shown already several times. The study presented here was based on patients in Taiwan. I have the following comments:
There are several types of pemphigus which vary in the severity of the disease. Please discuss the different types of pemphigus and indicate which types are considered in this paper, and how the disease and type were identified. Please also score the extent of skin and mucosal involvement of the disease on a scale of 0-3: 0 is no disease, 1 is minimal disease, 2 is moderate disease, and 3 is severe disease. For this scaling see Kumar et. al. Indian J. Dermatol. Venereol. Leprol. 2006, 72, 203-206. Please identify the treatment (steroids/immunosupressant/biologics etc.). This may be important as for instance high levels of steroids may influence depression. It is important to QUANTIFY depression, for instance by using the Hospital Anxiety Depression Scale (HADS). Furthermore, correlations with the Dermatology Life Quality Index scale and the Loneliness scale (Version 3 from UCLA) may be of interest. Quantifying depression may also be done using the Beck Depression Inventory.